# Fire risk to structures in California's Wildland-Urban Interface

Maryam Zamanialaei [1], Daniel San Martin[2], Maria Theodori[1], Dwi Marhaendro Jati Purnomo[1], Ali Tohidi [3], Chris Lautenberger[4], Yiren Qin[3], Arnaud Trouvé[3] & Michael Gollner [1] ✉

The destructive impacts of wildfires on people, property and the environment have dramatically increased, especially in the Wildland-Urban Interface (WUI) in California. In these areas structures are threatened by both approaching flames and lofted embers which spread fire into and within communities. While independent factors influencing structure fire protection are well known, their combined effects remain largely unquantified, limiting the accuracy of risk assessments and mitigation strategies. Here, we examine five major historical WUI fires—2017 Tubbs, 2017 Thomas, 2018 Camp, 2019 Kincade, and 2020 Glass Fires—utilizing machine learning (ML) analysis of on-the-ground post-fire data collection, remotely sensed data, and fire reconstruction modeling to assess patterns of structure loss and mitigation effectiveness. We show that the spacing between structures is a critical factor influencing fire risk, highlighting the importance of structure arrangement, while fire exposure, the ignition resistance (hardening) of structures, and clearing around structures (defensible space) work in combination to mediate fire risk. Utilizing an XGBoost classifier, structure survivability can be predicted to 82% accuracy. Results highlight the effectiveness of hardening and defensible space, with a hypothetical 52% reduction in losses. Our findings emphasize the need for community-level mitigation to reduce structure loss in future WUI fires.

Globally, the frequency, severity, and size of wildland fires has been increasing, resulting in extreme events that have led to dramatic losses in terms of people, property and the environment[1,2]. A majority of these impacts on people occur where houses and other urban development intermingle with undeveloped wildland vegetation, an area termed the WUI. This area has grown dramatically in recent years[3,4] with one-third of all new homes in the US built in the WUI[5]. The western United States has witnessed a 246% increase in structures lost to wildfires from 2010–2020 compared to the previous decade[6]. California, despite its long fire history, has experienced recent increases in the number of very large fires (over 100,000 ha) resulting in massive losses of lives and property[7]. Between 2013 and 2018, approximately 47,000 structures have been damaged or destroyed and 189 fatalities have been attributed to wildfires in California[8]. This increasing risk has consequences that jeopardize the economic stability, well-being of local residents, and the environment in affected communities[9].

Central to preventing future destruction has been the development of mitigation measures aimed at reducing the likelihood of ignition and spread in the WUI[10–14]. Improvements in building features and materials (hardening) and clearing surrounding vegetation and other flammable materials (defensible space) play important roles mitigating fire spread into the WUI[15–18] but differ in their characteristics

[1]Department of Mechanical Engineering, University of California, Berkeley, CA, USA. [2]Departamento de Informática, Universidad Técnica Federico Santa María, Valparaíso, Chile. [3]Department of Fire Protection Engineering, University of Maryland, College Park, MD, USA. [4]CloudFire Inc., Auburn, CA, USA. ✉e-mail: mgollner@berkeley.edu

because structures and vegetation have different heat release rates, durations of burning, and responses to external exposure including direct flame contact, radiation, and firebrands[19]. For instance, Ondei et al.[20] synthesize a zonation strategy for defensible space, focusing on removing dead vegetation within 1.5 meters of a house and managing fuel connectivity up to 30 meters. Similarly, studies like Carton et al.[21] stress the importance of fire-resistant construction, vegetation management, and the need for specific wildfire codes, particularly addressing the unique needs of Indigenous communities and heritage properties in Canada. While effective mitigation strategies have been developed based on past testing and investigations[22], their combined effectiveness under different exposure conditions is not yet known[5].

Previous geospatial studies have demonstrated the critical influence of spatial arrangement and biophysical factors[23–25], with defensible space around structures playing a substantial, albeit secondary, role[26,27]. The role of building materials has also been examined, revealing mixed findings[27–29]. For instance, Syphard and Keeley[28] found structural features like enclosed eaves and vent screens were crucial, while others (Price et al., Metz et al., Knapp et al.)[30–32] identified factors such as spacing and arrangement as more important, suggesting determinants of loss are often beyond homeowners' control. A later study of the Woolsey Fire suggests that proximity to destroyed structures and building materials, such as multi-pane windows and enclosed eaves, are key factors in determining survival.[27] Large structure loss datasets, such as those from the Camp fire, show that homes built before 1997 had markedly lower survival rates compared to those built after, underscoring the importance of construction standards[32]. The 2021 Marshall Fire also highlights the significance of neighborhood and parcel characteristics in housing survival, revealing the impact of jurisdictional differences in building codes and planning[31]. Collectively, these studies underscore that while defensible space is important, building features and surrounding vegetation, as well as proactive mitigation strategies, are critical to improving wildfire resilience.

Despite these advances, the majority of studies focus on single events, and lack a comprehensive quantitative analysis of how mitigation measures, such as home hardening and defensible space, interact and influence fire risk. In order to safeguard communities and stem the current trend of destruction, we must quantitatively understand how features influence fire risk to structures, particularly in relation to fire exposure, surrounding vegetation, the proximity of neighboring structures, and properties of the structures themselves. We hypothesize that the combined effects of structure hardening, defensible space, and structural separation can substantially reduce the risk of structure loss, with the most substantial benefits occurring when changes are made to both the structure itself and the surrounding vegetation. Furthermore, changes to individual structures may not be sufficient to reduce risk when structures are arranged at high density, requiring community-wide mitigation.

Here, we combine the largest existing structure loss database from California with simulated fire and ember exposure conditions to structures across multiple large-loss events, providing a methodology to quantify and compare the combined influence of exposure and mitigations such as defensible space and home hardening on fire risk. Unlike past studies, fire reconstruction modeling that includes urban fire spread is used to quantitatively estimate the effect of flame and ember exposure on structures. Geospatial assessments of vegetation surrounding structures are added using both LiDAR and visual imagery to assess the level of defensible space (vegetation) surrounding structures. The database is then fit using a multivariate analysis similar to refs. 27,31 that distinguishes between the interrelated effects of exposure, structure hardening, and defensible space. A parameter importance analysis reveals the strong role both structure separation and exposure play, distinguishing wildfire from other natural hazards that are not affected by neighboring conditions, highlighting the importance of a community approach to mitigation. The model

developed is strongly predictive when incorporating all the above features and is also used to assess the impact of recommended mitigation measures on homes. It is found that it is most impactful to make changes both to the structure itself and surrounding fuels, especially vegetation and other flammable materials within 1.5 m (5 ft) of the structure (zone 0) to achieve the maximum benefit.

## Results

In this study we took advantage of the Damage INSpection (DINS) Dataset collected by on-the-ground CAL FIRE crews from structures damaged, destroyed, or affected by wildfires in California during post-fire investigations between 2013–2022 (California Department of Forestry and Fire Protection (CAL FIRE))[33]. Figure 1 shows all fires in the DINS dataset between 2017–2022 as well as five of the largest loss fires in this dataset (2017 Tubbs and Thomas, 2018 Camp, 2019 Kincade, and 2020 Glass fires) selected for further analysis based on data availability and the number of structures exposed and destroyed. We combined records of damage state and building features from this dataset with remotely-sensed assessment of surrounding vegetation (akin to defensible space) and structure footprints (to assess building separation) of undamaged, damaged, and destroyed structures within the final fire perimeter (CAL FIRE Historic Fire Perimeters), including a 91 m (300 ft) buffer around any burned areas[34]. Post-fire reconstruction modeling was then used to add local fire exposure by both flames (flame length) and embers (ember load) to the dataset resulting in a more complete picture of fire exposure and effects.

Data from 5 selected fires were extracted from the overall DINS dataset (~90,000 structures) by combining/stacking the five fire datasets after preprocessing. Additional structures that were unburned but exposed to fire were added to the dataset (Tubbs ~14,000, Thomas ~ 6000, Camp ~24,000, Kincade ~2000, and Glass ~5000 structures). We employed a resampling process to balance the samples, resulting in a total of approximately 47,000 structures and 45,947 unique data points. We simplified the damaged, non-damaged, and destroyed classifications in the original DINS to a binary classification of Survived and Damaged categories because >90% of damaged structures are destroyed.

### Post-fire reconstruction

Five fires were reconstructed using a level-set model (ELMFIRE) that included both wildland[35,36] and urban fire spread[37] to re-create fire spread conditions and estimate critical missing exposure data (flame length and ember deposition) from these events. While reconstruction can never perfectly mirror on-the-ground conditions, these results provide reasonable estimates taking into account spatiotemporal variability in fuels, topography and weather. Figure 2 shows the modeled fires and resulting flame length (in meters) and ember load (in terms of number of embers deposited per meters squared). These are extracted adjacent to each of 47,000 structures in our dataset and distributions are shown in terms of flame and ember exposure as probability density functions (PDF) in Fig. 3. These distributions reveal a 27% and 39% overall decrease in exposure to flames and embers, respectively for structures that survived vs. those that were damaged. The decrease in exposure, however, is small in comparison to the difference in total number of structures destroyed and suggests that other features may play a role in determining which are more or less likely to survive.

### Feature contribution to structure loss

We applied an XGBoost[38] Classifier to our dataset and then utilized a SHapley Additive exPlanations (SHAP) model to explore the importance of various features on structure destruction. By looking at the stacked results from all 5 fires (Fig. 4), we found that structure density, which is determined by the distance between structures (SSD), is one of the most important features in structure destruction. The second

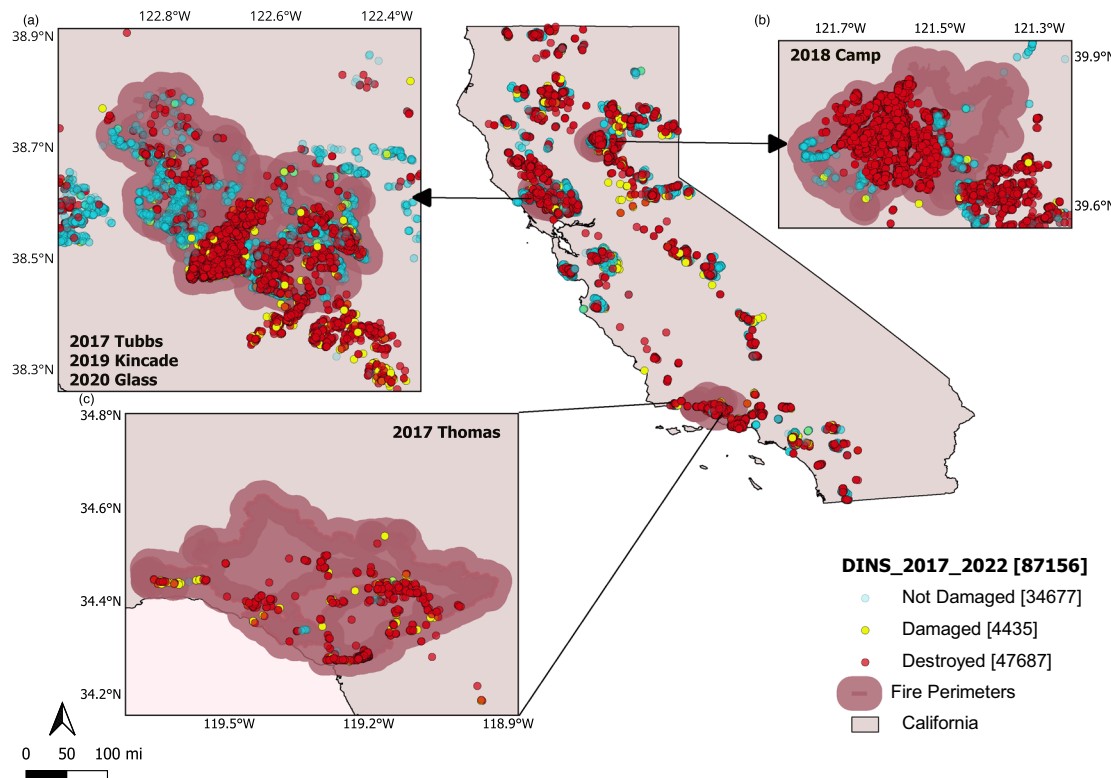

**Fig. 1 | Spatial distribution of damage to structures in California.** CAL FIRE Damage Inspection (DINS) data (2017–2022; *n* = 87,156) overlaid on fire perimeter polygons (semi-transparent rose shading) for five of the most destructive WUI fires before the 2024–2025 Los Angeles area fires. Insets show details for **a** 2017 Tubbs, 2019 Kincade and 2020 Glass fires (left panel), **b** 2018 Camp fire (upper right) and **c** 2017 Thomas fire (lower center). Symbols denote building damage state: cyan circles, Not Damaged (*n* = 34,677); yellow circles, Damaged (*n* = 4435); red circles, Destroyed (*n* = 47,687). The California state boundary is outlined in black. Coordinates are in degrees latitude and longitude; scale bar in miles. Map created using the Free and Open Source QGIS.

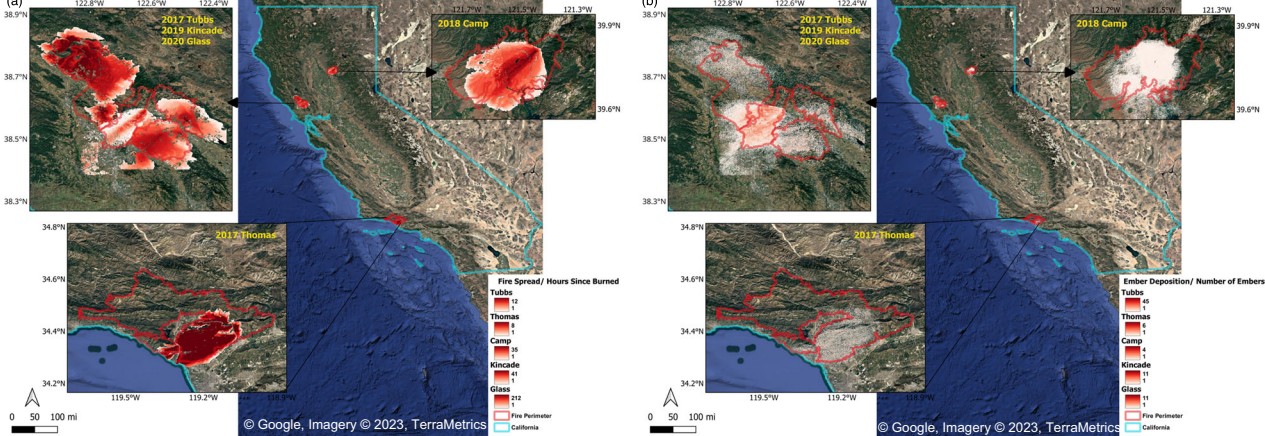

**Fig. 2 | Simulation of fire spread and ember deposition for the 5 fires, overlaid with fire perimeters. a** Fire spread (shaded based on time of arrival) illustrates fire progression in each event, while **b** ember deposition illustrates the large reach and stochastic nature of generated firebrands. Fire spread and ember deposition outputs generated using the ELMFIRE model with the HAMADA urban fire spread extension from elmfire.io (https://github.com/lautenberger/elmfire). In fire spread maps each pixel is colored by hours since burned or time of arrival (white = 1 h; dark red = maximum hours shown). Cumulative ember deposition used to show the average number of embers per cell (white = 1; dark red = highest mean). Fire perimeters (red outline) and the California state boundary (cyan outline) were overlaid in QGIS. Insets show details for 2017 Tubbs/2019 Kincade/2020 Glass (left), 2018 Camp (upper right) and 2017 Thomas (lower left).

most important contributor to the classification results from the XGBoost estimator was exterior siding, representative of the materials used in construction, followed by Year Built. Note, in the DINS database year built indicates the year that the primary structure in the parcel was constructed. Year built has therefore been identified as a confounding variable ultimately combining the effects from different parameters such as hardening (e.g., materials used for roof construction, eaves, vent screen, window pane, exterior siding), vegetation, and surrounding features (e.g., defensible space/vegetation separation distance and nearby structures/structure separation distance). Hence, considering Year Built as a single factor for determining vulnerability is inaccurate, and our results recommend adopting a holistic approach in such determinations. Results underscore the importance of hardening structures, structure density, and

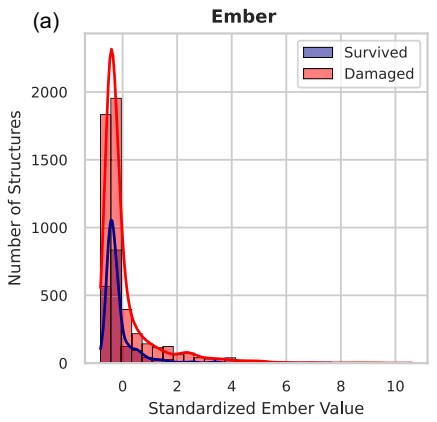

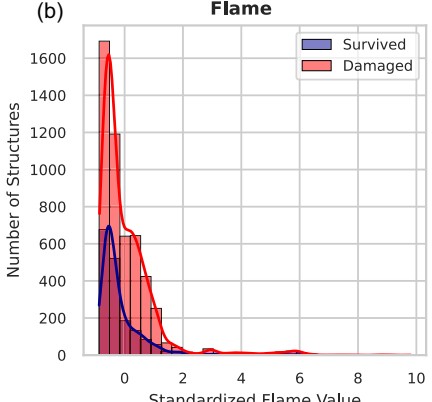

**Fig. 3 | Number distribution showing structure damage based on standardized flame and ember values.** Number distributions of structure damage versus standardized ember and flame values (*n* = 47,742). Histograms with overlaid kernel-density estimates display counts of surviving (blue) and destroyed (red) structures for **a** standardized embers and **b** standardized flame. These distributions highlight the pronounced effect of simulated ember and flame exposures on the destruction of structures in large WUI fires.

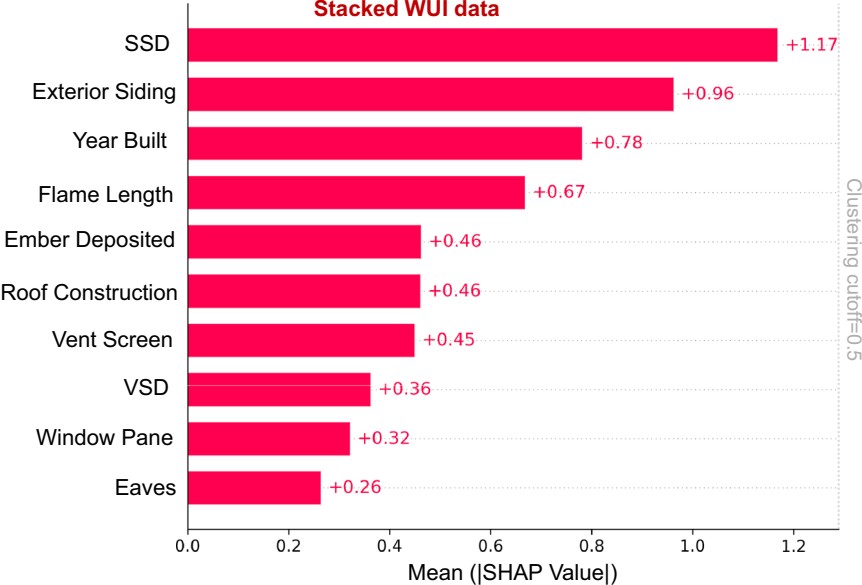

**Fig. 4 | SHAP aggregation results characterizing the contribution of features for the entire (stacked) WUI data from five fires.** Mean absolute SHAP values from an XGBoost classifier (*n* = 47,742 structures) trained on merged WUI data from five fires. Bars show the average |SHAP| for each predictor, ranked by contributions: structure separation distance (SSD), exterior siding, year built, flame length, ember deposition, roof construction, vent screen, vegetation separation distance (VSD), window pane and eaves. Higher |SHAP| indicates greater contribution to the model's prediction of structure destruction. The vertical dotted line marks the clustering cutoff (0.5) used to identify redundant features. SSD is the single strongest driver of predicted loss, followed by exterior siding and year built.

building arrangements in WUI areas to mitigate fire risk and potential destruction. These results are consistent with already-established engineering knowledge[22,32,39]. Furthermore, these insights are based on the available data rather than being drawn from direct experiments or detailed numerical simulations at the flame scale.

Exposure was still important in predicting damage from past WUI fires, specifically considering flame length and ember load derived from fire spread simulations. Flame length, which indicates the height and intensity of flames, can directly influence the severity of damage to structures and vegetation in its path. Ember load, representing the number and size of burning embers carried by the wind, also substantially contributes to the spread of the fire and subsequent structure loss, as these embers can ignite spot fires far beyond the main fire front. The intensity and reach of flames, as well as the quantity of embers, played pivotal roles in the extent of damage and structure loss observed in these fires.

SHAP values provide a unified measure of feature importance in a predictive model. Based on the evaluation metrics, the XGBoost model emerged as the most effective estimator, demonstrating superior skill in predicting losses. This finding is supported by its higher average SHAP values for key features compared to other models such as Logistic Regression and Random Forest. The average SHAP values for the XGBoost estimator revealed that certain features notably impacted the model's predictions. For instance, Structure Separation Distance (SSD) and flame length had positive average SHAP values of 0.090 and 0.051, respectively, underscoring the importance of building arrangements and fire behavior in risk assessment. Year Built showed an average SHAP value of −0.058, suggesting that newer structures might be associated with lower predicted losses, possibly due to improved building codes and materials.

We also broke down the feature importance results for each of the 5 individual fires assessed (Fig. 5), and found common features

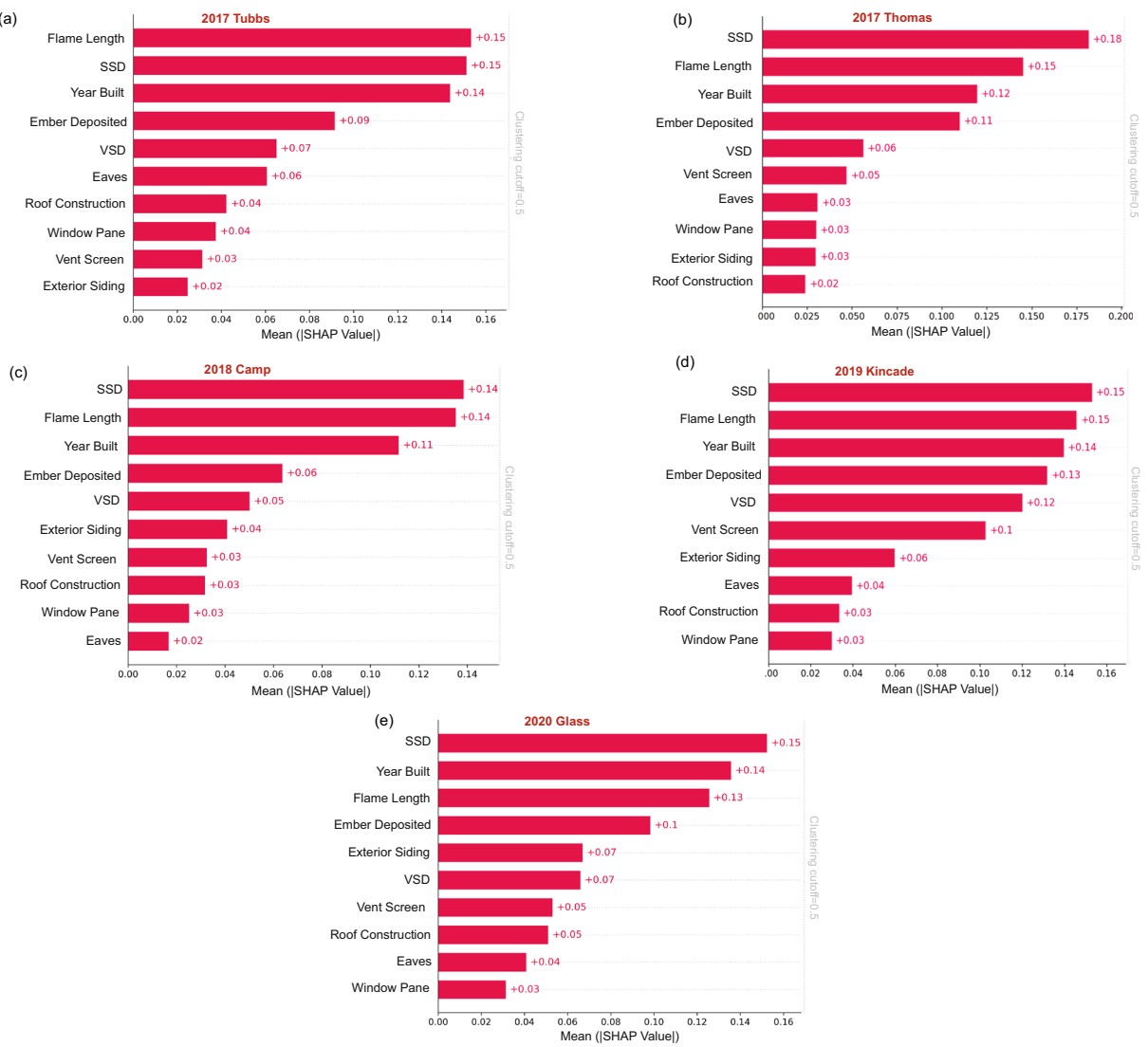

**Fig. 5 | SHAP aggregation results for characterizing the contribution of features for five individual fires.** Mean absolute SHAP values (mean |SHAP|) for the top ten predictors in each fire-specific XGBoost model. Sample sizes are: **a** 2017 Tubbs, n = 13,027; **b** 2017 Thomas, n = 5192; **c** 2018 Camp, n = 23,204; **d** 2019 Kincade, n = 1555; **e** 2020 Glass, n = 4768. Bars are ordered by decreasing mean | SHAP|; annotations show the numeric mean |SHAP| values. The vertical dotted line in each panel marks the hierarchical clustering cutoff (0.5).

of importance: SSD, flame length, and year built, but distinct differences were also revealed between individual fires. Structure separation distance (SSD) was the most important predictive feature in the 2017 Thomas, 2018 Camp, 2019 Kincade and 2020 Glass fires while flame length was the most predictive feature for the 2017 Tubbs fire. In fires that burn through densely populated areas, wildfires can transition into urban conflagrations that become dominated by structure-to-structure spread, which is most strongly influenced by SSD. The high density of structures in the Thomas and Camp fires in the cities of Paradise and Ventura (Butte and Ventura counties) therefore emphasize this mode of spread. In the Kincade and Glass fires, the clustering of structures in Geyserville (Sonoma County) and Deer Park (Napa County) contributed to the rapid urban spread of the fire. Flame Length substantially contributes to structure destruction in the Tubbs fire and is the second most important factor for the Thomas, Camp, and Kincade fires. In the Glass fire, it ranks third in importance, emphasizing the role of nearby buildings and surrounding fuels in spreading flames to structures. Year built, in conjunction with building characteristics (eaves, roof, vent, siding, window), underscores the significance of

home hardening in dense WUI areas, limiting fire spread and protecting structures from losses.

Figure 6 shows the distribution of four key features—SSD (structure separation distance), FLAME (flame length), YEAR BUILT (year primary structure on parcel was built), and EMBER (ember load)—across five fires: Tubbs, Camp, Glass, Kincade, and Thomas. Each panel represents one feature, displaying both the distribution of values through a violin plot (in light gray) and the mean values (in blue). The violin plots highlight the density of feature values, with wider sections indicating regions where values are more concentrated, allowing for a comprehensive comparison of how each feature behaves across different fires. For instance, the SSD and YEAR BUILT features show relatively wider distributions in the Glass fire, suggesting a broader range of structure separation distances and building years compared to other fires. Overlaid bar plots show the mean feature values for each fire, providing insight into the central tendencies. For example, the SSD and YEAR BUILT features have relatively higher means in the Camp and Glass fires, indicating greater separation distances and older structures on average in these regions. In contrast, the Kincade and Tubbs fires exhibit lower mean SSD values, suggesting tighter

structure spacing. The combination of these two plots makes it possible to assess not only the average feature importance (through the bar plots) but also the variation within each fire (through the violin plots).

## Damage prediction results

We applied a range of machine learning models, ultimately selecting an XGBoost Classifier as it was the most accurate to investigate the five large WUI fires in our dataset to predict structure survival during each fire. Linear models (logistic regression) have been used in the past[24,28,40] and achieved an accuracy to predict structure losses for our 5-fire database of 78%, In comparison, the CatBoost and Random Forest classifiers improved performance to 80% and 81% respectively, and the XGBoost classifier further increased accuracy to 82%. Beyond the numerical gains, XGBoost's ability to capture non-linear relationships and interactions among features, along with its robust regularization and efficient hyperparameter tuning, contributed to its superior overall performance. Consequently, after comparing related metrics—including accuracy, precision, AUC, recall and F1-Score—we selected XGBoost as the preferred model for our study.

A comparative analysis between the Logistic Regression, Random Forest, CatBoost, and XGBoost models is included in the "Methods" section and Supplementary Information (Supplementary Materials Figs. 1–3 and Supplementary Materials Tables 1–3) but importantly underscores the need for selecting an appropriate algorithm based on the specific characteristics of the dataset and outcome to be achieved. Additionally, we provide a ranking comparison that summarizes the predictive performance of our models across both the DINS dataset (2017–2022) and the combined dataset for five fires. Table 1 presents key performance metrics including Accuracy and AUC as well as the top three important features identified for each model, offering a comprehensive evaluation of each model's strengths and limitations. This ranking not only highlights the efficacy of our modeling approach but also provides valuable insights into the relative performance across datasets.

A confusion matrix is shown in Table 2 outlining the performance of our XGBoost classification algorithm breaking down predicted outcomes against actual results, delineating true positives (TP), true negatives (TN), false positives (FP), and false negatives (FN). The area under an ROC curve (AUC) is also shown as another measure to evaluate the model's overall performance alongside the accuracy (percentage of correct predictions the model makes). Overall the XGBoost Classifier has a high accuracy in predicting the occurrence of destruction for each of the 5 individual WUI fires, despite severe limitations in data coming from the aftermath of real destructive events. An accuracy threshold of 79% is achieved for each fire except for the Kincade fire (63%) which had a large number of missing values in the DINS inspection dataset and affected the results. The XGB classifier was also applied to the full DINS dataset (2017–2022) which incorporated the same preprocessing as the 5 fires but did not include exposure modeling values and defensible space, although it did incorporate all other analyses including structure spacing, and year built. An accuracy of 77% was achieved which demonstrates the flexibility and applicability of this model even when not all data can be accounted for.

Using our model we were able to examine various scenarios including home hardening and defensible space clearing to compare what changes in predicted structure loss and survivability might occur, in order to propose effective mitigation strategies. This is particularly important because structure density cannot be modified for existing structures (which make up more than 98% of the current housing stock[22,41]. We applied this to the 5-fire database we created. In the first scenario, which involved home hardening, we adjusted all hardening values in our dataset to fire-safe ones (e.g. non-flammable siding, fine mesh over vents, double paned windows, non-flammable roof, etc.) and applied the XGB model. This resulted in a 25% survival rate with

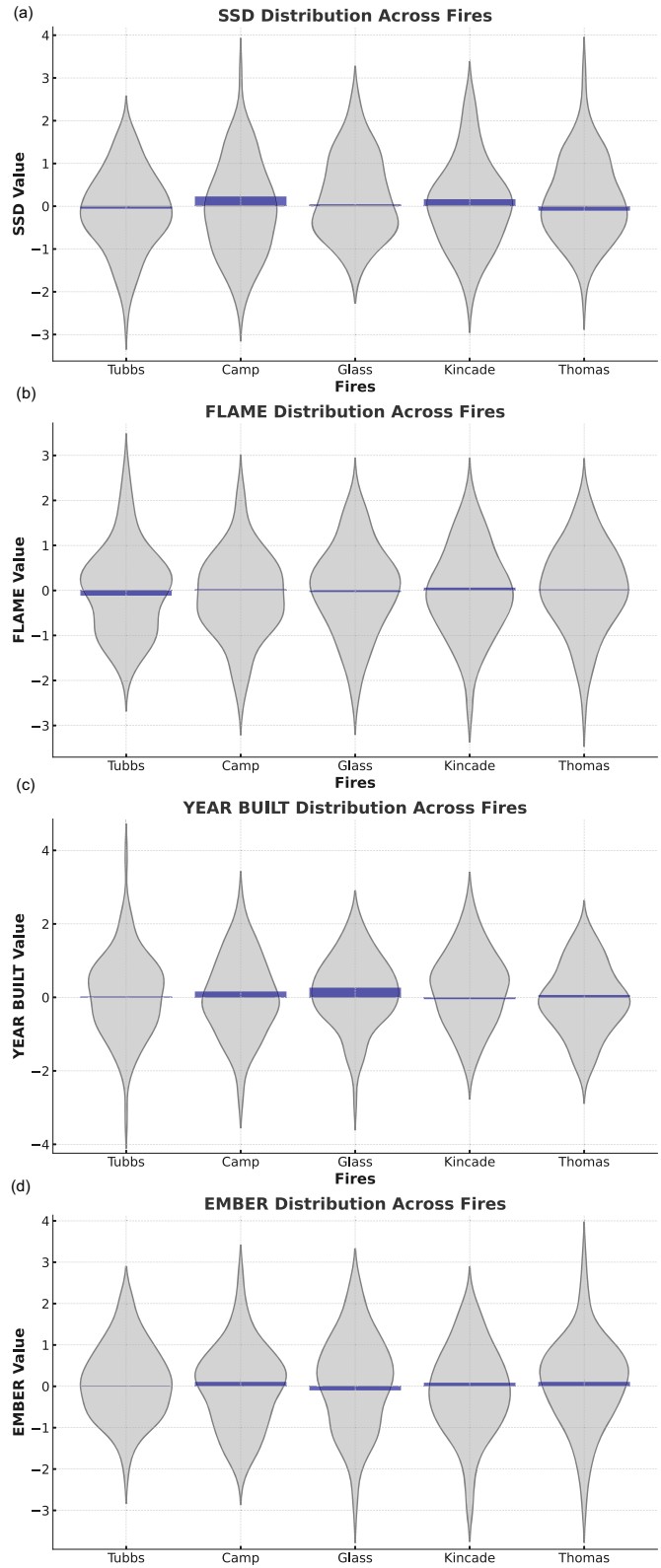

**Fig. 6 | The standardized distribution (gray) and mean values (blue) of important features across five fires.** Gray violin plots show the standardized distributions of **a** structure separation distance (SSD), **b** flame length, **c** year built and **d** ember deposition for each fire. A bold central line marks the median in each violin. Sample sizes are: Tubbs ($n = 13,027$), Camp ($n = 23,204$), Glass ($n = 4768$), Kincade ($n = 1555$) and Thomas ($n = 5192$).

**Table 1 | Comparative model performance ranking for DINS (2017–2022) and combined five fires datasets**

| Model | Dataset | Accuracy | AUC | Key features | | |
|---|---|---|---|---|---|---|
| Logistic Regression | DINS (2017-22) | 0.75 | 0.81 | Exterior Siding | Window Pane | Eaves |
| Random Forest | DINS (2017-22) | 0.75 | 0.82 | Exterior Siding | Year Built | SSD |
| XGBoost | DINS (2017-22) | 0.77 | 0.84 | Exterior Siding | Year Built | SSD |
| CatBoost | DINS (2017-22) | 0.75 | 0.80 | Exterior Siding | Year Built | SSD |
| Logistic Regression | 5 Fires Combined | 0.78 | 0.65 | Exterior Siding | Year Built | Vent Screen |
| Random Forest | 5 Fires Combined | 0.81 | 0.83 | Exterior Siding | SSD | Vent Screen |
| XGBoost | 5 Fires Combined | 0.82 | 0.83 | SSD | Exterior Siding | Year Built |
| CatBoost | 5 Fires Combined | 0.80 | 0.80 | SSD | Ember Deposited | Year Built |

**Table 2 | Results of XGBoost predictions on each test set with a resulting confusion matrix displaying model performance in terms of true positives (TP), true negatives (TN), false positives (FP), false negatives (FN), area under an ROC curve (AUC), and the percentage of correct predictions the model makes (Accuracy), the accuracy of positive predictions (Precision), model's ability to identify all positive instances (Recall), and the harmonic mean of precision and recall (F1 score)**

| WUI fire | TP | FP | TN | FN | AUC | Accuracy | Precision | Recall | F1-score |
|---|---|---|---|---|---|---|---|---|---|
| Tubbs | 1041 | 58 | 0 | 2 | 0.685 | 0.94 | 0.94 | 0.99 | 0.97 |
| Thomas | 147 | 25 | 31 | 21 | 0.808 | 0.79 | 0.85 | 0.87 | 0.86 |
| Camp | 3506 | 686 | 198 | 110 | 0.784 | 0.82 | 0.83 | 0.96 | 0.89 |
| Kincade | 27 | 58 | 124 | 27 | 0.635 | 0.63 | 0.31 | 0.50 | 0.38 |
| Glass | 151 | 58 | 496 | 106 | 0.841 | 0.79 | 0.72 | 0.58 | 0.64 |
| 5 Fires Combined | 4785 | 847 | 885 | 353 | 0.833 | 0.82 | 0.84 | 0.93 | 0.88 |
| All CA DINS (2017-22) | 5198 | 133 | 2998 | 1073 | 0.84 | 0.77 | 0.79 | 0.82 | 0.81 |

75% structure loss due to WUI fires (Supplementary Materials Fig. 6). Next, we combined home hardening with clearing defensible space in Zone 0 (0–5 feet; 0–1.5 meters), which effectively doubled the survival rate to 40% and reduced the loss rate to 60% (Supplementary Materials Fig. 7). Finally, we implemented an extreme mitigation scenario that included both home hardening and the clearing of defensible space in Zone 0 (0–5 feet; 0–1.5 meters) and Zone 1 (5–30 feet; 1.5–9 meters). This further increased the survival rate to 48% and reduced the structure loss to 52% (Supplementary Materials Fig. 8). Figure 7 shows the structure loss and survivability across various mitigation scenarios. It is crucial to acknowledge that these hypothetical scenarios did not incorporate the impact of suppression efforts or firefighting strategies, which could substantially influence the outcomes in real-world situations. Therefore, the results from this analysis should be considered as theoretical estimates, and the actual outcomes may differ considerably when these real-world mitigation strategies are applied.

## Discussion

Decades of research have shown the importance of ignition-resistant construction, defensible space, and the proximity of structures to one another[16,22,25,32,42]. The ranked importance and interplay between these mitigation measures has now been presented in this study, utilizing simulations to extract exposure conditions during different fires. The application of XGBoost and SHAP methods has illuminated the critical features contributing to structure destruction. Following investigations of the Camp fire by Maranghides et al. and Knapp et al.[22,32] Structure Separation Distance (SSD) arose as a key metric in characterizing the likelihood of loss for any particular structure during these large WUI fires. While smaller fires may occur through sparse housing arrangements, the majority of structure losses in California have occurred in large-loss fires in moderately dense (suburban) communities[28,32]. In these fires, the structures themselves become fuel and contribute to spread. These existing structures pose a unique challenge in hazard management—they are immobile. While these structures can be hardened, they cannot be readily removed or displaced like many other WUI fuels.

The analysis revealed the role of interactions among competing factors (like SSD, flame length, ember load, year built, and exterior siding) in influencing fire dynamics, showcasing that multiple features contribute simultaneously to fire risk to structures. Our results show that the most important factor that cannot be changed is the distance between structures, as conflagrations tend to consume a majority of houses in major fires. Nevertheless, there is still a substantial opportunity to enhance safety through effective mitigation measures such as hardening structures and establishing a defensible space. Mitigation measures on the structure (hardening) combined with removal of surrounding fuels in the area immediately adjacent to the structure (zone 0) has the potential to dramatically reduce losses in future fires. While applying these measures to any particular structure within a dense urban area makes little difference on the survivability of a single home, substantial reductions in losses are achievable when community-wide actions can be applied. This has been proposed in many studies and is a major tenant of community risk-reduction programs, however, it has not been shown to be effective before because previous studies focus more on individual structures. The effect of community risk reduction may have other benefits as well, with amplified effectiveness to responding fire crews[28]. When fewer homes ignite from embers or direct flame contact, less structure-to-structure spread results, and the fire service is freed up to focus on those structures that are most threatened. If arriving early in fire progression, it is possible that embers can be extinguished and the "disaster sequence" posited by Calkin et al.[10] can be disrupted and a smaller number of homes may be lost.

While hardening and defensible space actions may not alter the fundamental risks posed by structure proximity, they can still dramatically improve the survivability of buildings during wildfires. By examining individual fire cases, we can further tailor mitigation approaches to address specific vulnerabilities and enhance resilience against future wildfires. This is apparent when we see that the factors most correlated with structure destruction change for some fires, such as the Tubbs fire where flame length played a greater role than structure spacing overall. During this fire, sparser structures could

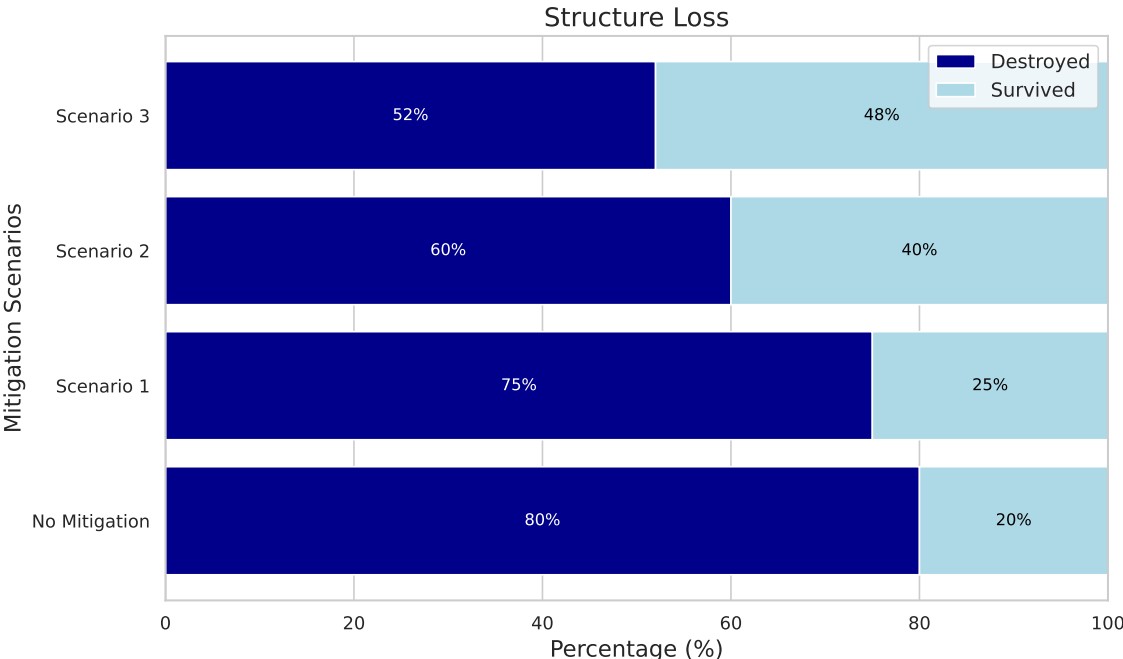

**Fig. 7 | Probabilities of predicted structure destruction under various mitigation scenarios.** Horizontal stacked bars show percentage of structures Destroyed (navy) versus Survived (pale blue) under various predicted mitigation scenarios: No Mitigation (baseline), Scenario 1 (home hardening only), Scenario 2 (home hardening and clearing Zone 0), and Scenario 3 (home hardening and clearing Zones 0 and 1). Percentage labels inside bars ($n = 47,742$).

potentially have benefited from additional fuel management to reduce fire exposure, although some areas, such as the Coffey Park community were dense and still dominated by structure separation. While siding materials were not critical to predict outcomes in any one fire, as an overall predictor they were very important, which speaks to the fact that clustered structures on each fire often had similar construction materials, year built, etc. so that those factors do not appear as important.

Overall, we have shown the potential to combine extensive on-the-ground post-fire data collection, analysis of remotely sensed data, and fire reconstruction modeling to better understand the complicated interactions between different features and structure survivability during wildfires. Despite the improvement of these data and modeling tools, we still have a dramatic lack of information before and during fires that is desperately in need of improvement. Pre-fire inspections are limited and were not available at scale to aid this study, therefore much of the data was collected after the fire and is not at the fine scale to distinguish all potential factors that play a role in destruction. For instance, year built is an inter-related term that also corresponds to materials used, construction type, building codes, etc. It is still useful because it is one of the most easily obtained factors for future analysis, but it makes it harder to distinguish between other factors. During the fires it would also have been useful to observe failure modes more directly, e.g. observation of what part of the exterior ignites by embers, structure-to-structure spread, etc. Still, this study provides a broader understanding useful to the field now and a framework for future data to be applied.

This analysis supports the effectiveness of home hardening and defensible space in reducing structure losses in the WUI, underscoring a need for more application in high-risk areas. Integrating fire risk assessments into land-use planning, including zoning regulations that incorporate risk maps to guide development away from high-exposure areas or require stricter building codes requiring defensible space and home hardening in communities that can be exposed to wildfires may help reduce losses. Similarly, policies that provide financial incentives, subsidies, or insurance benefits for retrofitting homes can also contribute to enhancing community resilience. These results can also be used to inform community outreach and education. Fire risk communication grounded in localized data can empower residents to take action and participate in preparedness efforts.

It is important to highlight potential limitations of the input data and methods used in this study. While reliance on post-fire damage inspection (DINS) assessments provides clear distinctions on the damage state of structures, there is inherent uncertainty on collected features that are difficult to determine in a post-damage state. Additionally, the DINS data may not capture the full range of potential factors that contribute to fire damage, such as local preparedness, fire suppression efforts, construction practices, etc., which could affect the generalizability of the findings to other regions or future fires. Remotely-sensed data, particularly for vegetation separation distance (VSD) is also subject to potential errors as small fuels that can contribute to local fire spread may not be visible at the resolution captured. The moisture content, species, and arrangement of vegetation and other flammable materials may also influence the effectiveness of defensible space but can't be captured through remote sensing modalities. Estimates of fire and ember exposure from fire reconstruction modeling are also subject to potential errors from deviations in fuel or weather input data and the empirical nature of the model itself. Machine Learning models are also limited in their reliance only on provided data. Efforts have been taken to minimize these potential sources of error, including conducting a comprehensive sensitivity analysis with perturbations, an assessment of the trained ML model to systematically assess how variations in key parameters influence our predictions, and use of a validation dataset to assess the performance of the model.

## Methods

We primarily relied on a modified database from five selected fires that includes more than 47,000 structures with two broad damage states: "Survived" and "Destroyed", and five detailed damage states: "Destroyed (>50%)", Damaged ("Major (26–50%)", "Minor (10–25%)", "Affected (1–9%)"), "No Damage". The CAL FIRE Damage INSpection Program (DINS) was founded with the goal to collect data on damaged, destroyed, and unburned structures during and immediately after fire

events to assist in the recovery process, and to provide local governments and scientists information for analyzing why some structures burned and why some survived[43]. Through a public records request, we acquired DINS data for more than 90,000 structures that survived, were damaged, or were destroyed across all California wildfires from 2013–2022, making this potentially the largest combined dataset of its sort. We then incorporated risk factors associated with structure destruction by wildfires to the DINS data to gain a deeper understanding of WUI destruction. These factors include structure density, building materials, year built, defensible space, and exposures to structures (fire intensity and ember). We employed several Machine Learning (ML) techniques to identify and highlight the important features in our WUI data. These techniques included feature selection, feature engineering, and model interpretation methods to ensure we could pinpoint the most influential variables influencing our results. To enhance the performance of the ML model in this study, we implemented a range of data preprocessing techniques such as data cleaning, normalization, and encoding. These preprocessing steps were crucial for improving model accuracy, reducing noise, and ensuring the robustness of our findings. By meticulously preparing the data, we ensured that the ML model could effectively learn and make accurate predictions from our complex WUI dataset. We opted for the XGBoost (eXtreme Gradient Boosting) algorithm for our ML model due to its superior performance over other methods on our dataset. We also leveraged the SHAP (SHapley Additive exPlanations) model, which provides a nuanced understanding of each column's contribution to the overall predictive outcome. This technique allowed for a comprehensive assessment of the importance of variables within the dataset, enhancing the robustness and reliability of our analysis. The results of Confusion Matrices and Receiver operating characteristics (ROC) Curves, in addition to an advanced computational framework, allowed us to delve into the intricacies of the dataset, capturing complex relationships and patterns that might not be discernible through conventional methods. Our evaluation extended beyond a generalized assessment, as we calculated the accuracy and sensitivity metrics for each individual fire and aggregated the results to encompass all structures within the damage dataset. This meticulous analysis not only provided insights into the predictive performance of our model on a per-fire basis but also yielded a comprehensive understanding of its effectiveness across the entire spectrum of structures in the damage data.

## Risk factors to structures from wildfires in the WUI
The methodology for integrating risk factors related to structure destruction builds upon the combination of on-the-ground data with fire modeling reconstructions by Hakes and Theodori et al.[34] for community-level risk assessment for the Tubbs fire, which includes:

*Structure spacing* which represents "Structure Separation Distance (SSD)". We employed the Microsoft Maps dataset (available at https://github.com/microsoft/USBuildingFootprints), which encompasses open building footprints datasets for entire counties in the United States. This dataset comprises 129,591,852 computer-generated building footprints. Additionally, we utilized QGIS software to access geospatial data concerning urban infrastructure, building locations, and their spatial interconnections.

The *year built* refers to the year in which the primary structure on a parcel of land was constructed. In the context of analyzing the impact of WUI fires, the Year Built variable is important because the age of a structure can influence its susceptibility to fire damage. Furthermore, it acts as a confounding variable that can affect both the building features and the extent of damage.

Concerning fire safety in *building construction materials*, numerous in-depth studies have been carried out through meticulously planned laboratory tests[18,44]. Despite the solid laboratory evidence, few empirical studies have documented building characteristics associated

with structure loss in real wildfire situations[28]. In this study building characteristics include eaves, vent screens, exterior siding, roof construction, and window panes.

In terms of *defensible space*, which is representing in this study as "Vegetation Separation Distance (VSD)", the state of California requires fire-exposed homeowners to create a minimum of 30 m (100 ft) of defensible space around structures, and some localities are beginning to require at least 60 m (200 ft) in certain circumstances[26]. We established three categories for the Vegetation Separation Distance (VSD): Zone0, which comprises the initial five feet from the building or "0–5"; Zone1, encompassing the area within 30 feet of the building or "5–30"; and Zone2, extending to within 100 feet of the building or "30–100" (CAL FIRE DSpace: https://www.fire.ca.gov/dspace). Remote sensing techniques were utilized to analyze the density and distribution of vegetation in the WUI regions and urban settings, extracting valuable insights from the aerial and satellite imagery and LiDAR data. The publicly available datasets (including countywide LiDAR data and a fine scale vegetation and habitat map) which were produced by the Sonoma County Agricultural Preservation and Open Space District and the Sonoma County Water Agency, provide an accurate, up-to-date inventory of the county's landscape features, ecological communities and habitats (Sonoma County Vegetation Map: https://sonomavegmap.org/).

*Exposures* including fire intensity (flame length) and firebrand (ember load). Houses are destroyed during wildfires when exposed to flames in adjacent fuel, radiant heat from nearby fuel (≤40 m)[16], or airborne embers and firebrands originating in nearby and distant fuel (typically < 10 km)[45,46]. In this study, we used the Eulerian Level set Model of FIRE spread, ELMFIRE, an operational fire behavior and spread simulation tool[35] for its additional capability in simulating ember deposition of multiple embers and its implementation of Monte Carlo analysis[36] to capture the stochasticity and uncertainty inherent in wildland fire modeling. We used and modified the semi-physical model of [36,47] to include urban fire spread by using the empirical approach of HAMADA[37].

## Data preprocessing
To predict the damage for any of the fire datasets, the dataset was divided into the target variable or y, and all the other features as inputs or X. A stratified split was executed based on "y" values, allocating 80% of the data for training purposes and reserving the remaining 20% for the testing set. This stratified approach ensured that the class proportions in the target variable were similar in both subsets, minimizing the risk of bias due to imbalanced classes. By preserving the target class distribution, this partitioning strategy not only improved the model's ability to generalize but also provided a more accurate and reliable performance evaluation when tested on unseen data. Additionally, the use of a fixed random_state ensured that the split was reproducible, allowing for consistent model training and evaluation across different iterations. As part of the model training process, we utilized GridSearchCV for hyperparameter tuning across several models, including Logistic Regression, Random Forest, and XGBoost. During the grid search, k-fold cross-validation (with cv_k_folds set to 10) was employed to evaluate the models, ensuring robust validation and mitigating overfitting. In the cross-validation process, the data was split into k-folds, where each fold served as the validation set once, while the remaining k-1 folds were used for training. This allowed the grid search to identify the optimal set of hyperparameters based on performance metrics, such as accuracy and F-beta scores. After selecting the best hyperparameters, the model was refitted on the entire training set, ensuring that the final model was well-tuned for testing.

To address the noteworthy variations in the scales of the model inputs, a vital preprocessing step was implemented prior to model training. Using the scikit-learn package[48], we first designed imputation

strategies through IterativeImputer to handle missing values. These strategies were trained on the training set and then applied to both the training and test sets. The imputation strategy was tailored for each feature in stacked WUI data and for each wildfire case. For example, Roof Construction (19,318 non-null), Eaves (19,318 non-null), Vent Screen (19,318 non-null), Exterior Siding (19,318 non-null), Window Pane (19,318 non-null), VSD (3504 non-null), and Year Built (22,501 non-null) were imputed using a nearest neighbor approach. For Year Built in individual fire cases, either nearest neighbor imputation or a median-based strategy was adopted, whereas numerical features like Embers (11,549 non-null), and Flame length (14,578 non-null) were aggregated (e.g., using the mean or median, potentially augmented by k-nearest neighbors) to fill in missing values. In our approach, we incorporated a spatial clustering technique that utilizes proximity-based methods for data imputation. Specifically, we leveraged Haversine Distance and Pairwise Distance metrics in UTM coordinates to cluster data points based on their geographic proximity. This spatial clustering approach ensures that similar locations, defined by latitude and longitude, are treated consistently when imputing missing values. By considering spatial proximity, we make the assumption that nearby data points are likely to share similar attributes, enhancing the robustness of the imputation process. Next, we normalized the numerical variables using StandardScaler, ensuring that they were on a similar scale, which helps in the convergence and performance of various models. Additionally, we conducted OneHotEncoding and Label Encoding on categorical variables using OneHotEncoder and LabelEncoder from scikit-learn to convert them into a numerical format that can be understood by the models. Class balance is achieved through the binarization of different labels/classes with damaged and not damaged/survived. This approach is essential, particularly in scenarios where certain damage classes may be underrepresented. This preprocessing pipeline allowed us to use a variety of models on the dataset, ensuring compatibility and enhancing the overall performance of the models.

In essence, this procedure, encompassing data categorization, stratified splitting, imputation, standard scaling, OneHotEncoding/ Label Encoding, and resampling, laid the foundation for a robust and unbiased evaluation of the model's predictive capabilities regarding fire damage across diverse datasets.

## Machine learning techniques

Machine learning (ML) methods have recently been applied to wildland fire[49] and present an ideal platform for WUI fires as interactions between competing factors can be fit and modeled. In this work, we employed both regression and classification ML techniques to our combined dataset resulting in a predictive model for structure destruction based on home hardening (roof, siding, vents, eaves, window, year built), vegetation separation (defensible space and surrounding), exposure metrics (flames and embers), and structure spacing. The XGBoost (eXtreme Gradient Boosting) machine learning algorithm was chosen as it outperformed other methods on our dataset. The model hyper parameters were tuned using RandomizedSearchCV, which was employed to perform a randomized search over a predefined parameter grid. This approach was used because of the large number of parameters in the XGBoost model. Hyper parameter selection is performed using the best result in terms of the following classification metrics: F-beta[50], F1-Score[50], accuracy[51], balanced accuracy[52] and precision-recall scores[53]. The F-beta score is used to balance precision and recall, with the beta parameter allowing for tuning the model's sensitivity to false positives and false negatives. Finally, feature importance with SHAP aggregation analysis was utilized to quantify the contribution of each feature to the target variable. A higher feature importance score indicates that the feature has a greater influence on the model's prediction[54]. The SHAP model connects optimal credit allocation with local explanations using the classic

Shapley values from game theory and their related extensions[55]. This was then applied to a unified framework for interpreting predictions to explain the output of any machine learning model.

## Classifiers

We employed several classification models, including Logistic Regression and Random Forest[48], and Gradient Boosting based XGBoost[56] since there is another method called Gradient Boosting Machine other than Extreme Gradient Boosting Machines (XGBoost). Each of these models offers distinct advantages and methodologies for analyzing feature importance.

Logistic Regression is a generalized linear model used for classification problems[57] and we use it as a base model to compare with more complex models. The second model used in this work is the Random Forest. Random Forests are a technique in ensemble learning utilized for tasks such as classification and regression. During the training, several decision trees are built. In classification, the random forest outputs the class chosen by the majority of trees[58]. CatBoost employs an ordered boosting technique to minimize target leakage from categorical features, often leading to robust performance even with limited parameter tuning[56]. While CatBoost can seamlessly integrate categorical data with minimal preprocessing and achieve competitive performance on binary classification tasks, logistic regression, random forest, and XGBoost typically require more elaborate feature engineering and preprocessing, which in turn can influence both model performance and the interpretability of sensitivity analyses such as those based on SHAP values. Finally, Gradient Boosting (GB) is a method in machine learning that employs boosting within a functional framework. The XGBoost (eXtreme Gradient Boosting) is a GB implementation that has been used as it outperformed other methods on our dataset. XGBoost is often preferable for developing predictive models for large datasets due to its accuracy, efficiency, and adaptability[38]. Furthermore, XGBoost is a robust algorithm for both classification and regression problems. Due to its strengths in model prediction, XGBoost can be utilized for damage assessment to create predictive models for structure destruction. The SHAP analysis results for all four models are provided in the Supplementary Materials (Supplementary Figs. 1–3). These figures offer a detailed breakdown of how each feature contributes to the predictions across models, enhancing the interpretability of our findings and complementing the results discussed in the main text.

## Feature contribution through SHAP analysis

While machine learning (ML) models are increasingly used due to their high predictive power, their use in understanding the data-generating process (DGP) is limited. Understanding the DGP requires insights into feature-target associations, which many ML models cannot directly provide, due to their lack of understanding causal effects. Feature importance (FI) methods provide useful insights into the DGP under certain conditions[59]. Furthermore, SHAP (SHapley Additive exPlanations) is a unified framework for interpreting machine learning models based on cooperative game theory[55]. It assigns each feature an importance value for a particular prediction by computing the contribution of each feature to the prediction, averaging over all possible combinations of features. This approach ensures consistency and local accuracy, providing insights into how different features influence model predictions. SHAP values can explain individual predictions and provide a global understanding of the model's behavior, making it a valuable tool for model interpretability in research[54]. SHAP can be considered a form of in-sample sensitivity analysis because it assesses how changing a feature or a subset of features affects the model's output. It evaluates the impact of including or excluding a feature and identifies which features contribute most to the predictions[60]. We utilized SHAP interpretation analysis of feature importance to identify and understand the key factors driving structure destruction in WUI

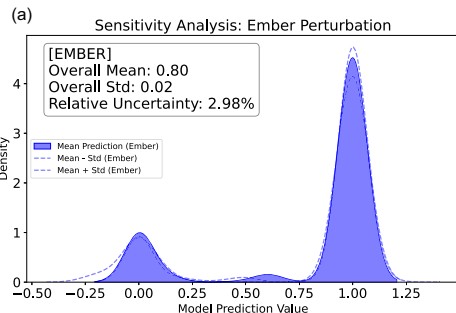

**Fig. 8 | Sensitivity analysis with respect to ember and flame exposure with perturbations.** A sensitivity analysis is shown, performed by perturbing two key exposure inputs—ember deposition and flame length—using 100 ensemble outputs from the ELMFIRE spread model (HAMADA extension) for each fire, while holding all other predictors constant. For each test sample ($n = 47,742$), the model's predicted survival probability was computed across ensembles to yield a mean prediction (solid fill) and its uncertainty (shaded region = ± 1 standard deviation). **a** Ember perturbation: blue fill shows the kernel density of mean predictions under varied ember load; dashed blue lines mark mean ± std. Inset reports overall mean, standard deviation and relative uncertainty (%). **b** Flame perturbation: green fill shows the kernel density of predictions under varied flame length; dashed green lines mark mean ± std. Inset reports overall mean, standard deviation and relative uncertainty (%). This framework quantifies the influence of non-linear interactions and input variability on the binary damage classification (0 = not damaged, 1 = damaged), reveals emergent intermediate prediction modes, and offers both local and global insights into model behavior under uncertainty.

fires. In this study, we opted for SHAP (SHapley Additive exPlanations) as a model-agnostic tool because its values not only quantify the magnitude and direction of each feature's contribution, but also capture complex non-linear interactions between variables[55]. This provides both local and global insights that are critical for understanding the multifaceted nature of fire damage. For example, SHAP allowed us to reveal how features such as SSD, ember exposure, and flame length interact in non-linear ways that traditional importance measures might overlook. Ultimately, the detailed and context-specific information provided by SHAP helped us interpret the predictive factors driving structural vulnerability, reinforcing the robustness of our findings.

### Sensitivity analysis for the machine learning model
We developed a comprehensive sensitivity analysis framework to assess how variability in key input features from the exposure model (ember load and flame length) affects our model predictions. For each of the five fires, ensemble outputs from the WUI fire spread model were used to perturb the "ember load" and "flame length" variables while keeping other inputs fixed. By aggregating the model outputs from these multiple ensemble runs, we computed the mean predictions and corresponding uncertainties for each test sample. This approach allowed us to quantify the impact of non-linear interactions and input variability on the final predictions, offering both local and global insights into model performance.

Visualizations, such as kernel density estimation (KDE) plots, clearly illustrate the distribution and variability of the predictions across the test samples (Fig. 8). The shaded regions represent the uncertainty around the mean predictions for both ember and flame perturbations, with the respective overall mean, standard deviation, and relative uncertainty values indicated within the plots. These distributions provide a clear view of the uncertainty and variability in the model's response to perturbations in ember load and flame length. Additionally, SHAP analysis was employed to further interpret the contributions of each feature, enhancing our understanding of the model's behavior under different exposure conditions. This sensitivity analysis not only characterizes the associated uncertainties related to flame and ember in the model but also suggests that the machine learning estimator, XGBoost, has learned an underlying understanding of the problem implying intermediary outcomes other than damaged and survived are possible in the dataset; see the emerged middle class distributions in Fig. 8. Additionally, it helped us gain insights into the physical factors influencing damage, as it highlights the non-binary classifications for the damage classes, offering a more nuanced understanding of the damage severity.

### Confusion matrix and ROC curve for predictions
A confusion matrix summarizes the classification performance of a classifier with respect to some test data. It is a two-dimensional matrix, indexed in one dimension by the true class of an object and in the other by the class that the classifier assigns[61]. Receiver operating characteristics (ROC) graphs are useful for organizing classifiers and visualizing their performance. A receiver operating characteristics (ROC) graph is a technique for visualizing, organizing and selecting classifiers based on their performance[62]. We investigated the five large WUI fires in our dataset to predict structure survival during each fire by understanding the model's accuracy, and other key performance metrics. By analyzing the confusion matrices and ROC curves for each fire event, we were able to identify patterns and discrepancies in model performance, leading to a better understanding of the factors influencing structure survival in large WUI fires.

### Reporting summary
Further information on research design is available in the Nature Portfolio Reporting Summary linked to this article.

## Data availability
The datasets generated during and/or analyzed during the current study are available in the [DINS_data_analysis] repository, [https://github.com/berkeley-firelab/DINS_data_analysis]; [https://doi.org/10.5281/zenodo.15776778][63].

## Code availability
The code used to conduct the analysis in this study is available in the [DINS_data_analysis] repository at [https://github.com/berkeley-firelab/DINS_data_analysis]; [https://doi.org/10.5281/zenodo.15776778][63].

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

## Acknowledgements

We would like to express our deepest gratitude to Matt Lee, Dave Shew, Dave Sapsis, Steve Hawks, and William Brewer from CAL FIRE for their invaluable contributions, guidance, and expertise throughout this study. Funding for this project was provided by the California Department of Forestry and Fire Protection's Forest Health Program as part of the California Climate Investments Program, grant 8GG21815. Additional funding by the Gordon and Betty Moore Foundation (11999), and the National Science Foundation (NSF) (1854952) also supported this work.

## Author contributions

M. Zamanialaei performed the conceptualization, data curation, data analysis, and wrote the original draft. D. San Martin carried out the conceptualization, data analysis, and methodology. M. Theodori performed the data curation, conceptualization, and methodology. D. Purnomo contributed to the conceptualization, and reviewing and editing. A. Tohidi conducted data analysis, visualization, methodology, and reviewing and editing. CH. Lautenberger contributed to the methodology and reviewing and editing. Y. Qin contributed to the methodology and reviewing and editing. A. Trouvé contributed to the methodology and reviewing and editing. M. Gollner supervised the work, contributed to conceptualization, and reviewed and edited the writing.

## Competing interests

The authors declare no competing interests.
