## [Transparent Peer Review file · Nature Communications]

Fire Risk to Structures in California's Wildland-Urban Interface

Corresponding Author: Professor Michael Gollner

Version 0:

Reviewer comments:

Reviewer #1

(Remarks to the Author)

Thank you for the opportunity to review the manuscript titled "Isolating the Primary Drivers of Fire Risk to Structures in WUI Regions in California". This study explores the critical factors influencing structure loss during wildland-urban interface (WUI) fires, particularly in California. The authors provide valuable insights into how home hardening, defensible space, exposure to flames and embers, and structure separation contribute to fire risk. By integrating machine learning models with exposure data, the study achieves a predictive accuracy of up to 82%, offering actionable data-driven strategies for fire mitigation. While this research presents compelling findings, several areas require significant revision to align with the standards of Nature Communications and enhance the manuscript's clarity, rigor, and relevance.

The abstract is well-structured but should be refined to explicitly highlight the context, objectives, methods, results, and conclusions. A concise abstract is essential for effectively communicating the study's contributions to a broad readership.

Additionally, the language should be made more engaging to encourage downloads and citations.

A well-defined research problem is crucial for a strong manuscript, and while this study addresses an important issue, the problem statement could be more clearly articulated. The introduction should explicitly state the gap in existing research and how this study fills that gap. Clearly outlining the research questions and hypotheses will provide readers with a compelling rationale for the study and its broader significance.

The discussion section should include a more thorough examination of the study's limitations. For example, the reliance on post-fire damage assessments, potential biases in the dataset, and limitations of the machine learning approach should be acknowledged. Furthermore, the manuscript would benefit from a stronger discussion of policy implications. Given the increasing severity of wildfires in WUI regions, how can these findings inform fire safety regulations, urban planning, or emergency response strategies? Adding specific recommendations will significantly enhance the real-world applicability of the research.

The manuscript presents noteworthy results. The finding that structure separation and exposure to embers and flames are the strongest predictors of structure loss is particularly significant. The results highlight the crucial role of urban conflagrations in fire spread, distinguishing them from smaller wildland fires. The study also provides compelling evidence that home hardening and defensible space—particularly within five feet of a structure—can reduce hypothetical structure losses by 52%. This reinforces the importance of mitigation strategies tailored to WUI communities.

The study is highly relevant to wildfire risk management and has potential applications in related fields such as urban planning, forestry, and disaster resilience. When compared to existing literature, the research aligns with previous studies emphasizing the role of home hardening and defensible space. However, some findings, particularly regarding the dominance of structure separation over other factors, challenge traditional assumptions. Expanding the literature review to provide more context for this finding would strengthen the discussion.

While the data analysis is robust, the study could benefit from additional validation. The machine learning models used are effective, but the conclusions regarding mitigation strategies should be carefully framed to acknowledge that real-world application may vary. Additional case studies or sensitivity analyses would help confirm the generalizability of the results.

The methodology is sound and meets expected standards in the field, particularly in the use of geospatial assessments, machine learning techniques, and predictive modeling. However, more detail is needed to ensure full reproducibility. The manuscript should provide clearer descriptions of data preprocessing steps, variable selection criteria, and model training procedures. Transparency in these areas will allow other researchers to replicate and build upon the findings.

In addition to these points, a few formatting and structural improvements are needed. The journal recommends keeping the main text (Introduction, Results, and Discussion) under 5,000 words and limiting the number of figures and tables to 10.

Ensuring adherence to these guidelines will improve the manuscript's chances of acceptance. The reference list should also

be expanded to include 60–70 citations, prioritizing recent and globally relevant studies to strengthen the paper's scholarly foundation.

This manuscript offers valuable insights into wildfire risk and mitigation, making it a strong candidate for publication. However, significant revisions are required to refine the abstract, enhance the discussion of policy implications, address limitations, and ensure methodological clarity.

Reviewer #2

(Remarks to the Author)

Hello! It was a pleasure to read your paper. It's good to see new research and methods reinforce previous results. The writing is clear, concise, and effective.

Noteworthy results are that:

- Several factors are at play during a fire event that affect the outcome of structure loss (previous research had similar conclusions, but this is a new approach and reinforces previous results).
- there is still opportunity to mitigate fire risk.
- Community-wide actions are important.
- Structure Separation Distance is a key metric when characterizing likelihood of loss.

The methods are sound as far as my understanding and there is enough detail in the methods section for the work to be reproduced.

Minor correction(s): I noticed that one of your references is missing from the reference's list (Mockrin et al., 2023); This is biased on my end because I was the author, but on the first paragraph of the Discussion (lines 2/3), my research from 2016 also said that the spatial arrangement of building plays an important role. That paper is already in the reference list, but I confess I feel proud to see other researchers, with different methods, came to similar conclusions. If you agree with my take, would you be willing to add my paper "The relative impacts of vegetation, topography and spatial arrangement on building loss to wildfires in case studies of California and Colorado"?

I recommend publishing this paper.

Thank you,
Patricia Alexandre.

Reviewer #3

(Remarks to the Author)

The effect of the Wildland-Urban Interface (WUI) was investigated in this study using extensive datasets. The authors combined California's largest existing structure loss database with simulated fire and ember exposure conditions to structures across multiple large-loss events, comparing the effects of structure hardening and nearby defensible space. Fire reconstruction modeling including urban fire spread was implemented using Eulerian Level Set Model of FIRE (ELMFIRE). Three machine learning algorithms (CatBoost algorithm was mentioned on page 10, but nothing was mentioned in the Methods and Results section. Therefore, it was ignored in my report), including logistic regression, random forest, and XGBoost algorithms. Logistic regression method is one of the simplest methods applied in the literature as a benchmark method, but its performance is usually poorer than advanced machine learning and deep learning algorithms. Three scenarios were conducted to measure the effect of mitigation measures and it reported that a significant reduction in the loss can be achieved with these scenarios. I think the paper has some novelty and may deserve to be published. However, there are some issues that the authors must consider to revise in terms of technical soundness and compliance with journal style.

-Random forest has been a popular ML algorithm, and it is used in almost all fields, with high level of performance reported. One of the most superior characteristics of the Random Forest algorithm is that it can be directly used for feature importance estimation using its Mean Decrease Impurity (MDI) or Gini importance. In this study, the performance of the random forest method was not provided. In the first paragraph of the Damage Prediction Results section, prediction performances of logistic regression and XGBoost were given as estimated accuracies of 78% and 82%. In the next paragraph of that section, although the statement "A comparative analysis between the Logistic Regression, Random Forest, XGBoost and CatBoost models is included in the methods section, but importantly underscores the need for selecting an appropriate algorithm based on the specific characteristics of the dataset and outcome to be achieved" was given, no result was provided for random forest (also CatBoost as mentioned above) in the main document and the supplementary document. This is one of the main results that must have been given, and a comparative discussion should be provided. The authors should discuss the similarities and differences between the SHAP graph of Random Forest and its Gini index results (for the entire datasets as given in Figure 4). SHAP is a model-agnostic approach and can be applied to any method. However, feature importances of random forest are automatically produced by the model. SHAP findings for feature importances may be different, which requires a short discussion (this is important to prove the trustworthiness of the SHAP method as an explainability model).

A similar problem exists in the interpretation of the logistic regression results in that SSD, which is the most important feature in the XGBoost model, was the least effective feature. The accuracy difference was about 4%, and such a difference was reported in feature importances. Please make some comments on the results, by giving possible reasons.

- In the title instead of using the term "Isolating", I suggest the authors should prefer another term (e.g., Determining, Uncovering, Identifying).
- Authors should mention the name of the methods and the scenarios in the abstract section. In its current form, it is too generic, not including the details of the study and important results achieved in this study.
- Please follow the instructions for authors on how to cite articles in the main text (numbering) and section heading. I think the section called Main should be revised as Introduction.
- I am confused about the datasets, more specifically the features used in the analysis. On page 1 (where the authors start to explain the data in their study)", it was stated that " Geospatial assessments of vegetation surrounding structures are added using both LiDAR and visual imagery to assess the level of defensible space (vegetation) surrounding structures." Did you use LiDAR and visual imagery data in this study to prepare your dataset (in SHAP graphs there are totally 10 features or factors considered)? Please clarify this issue.
- Please give some information on the fire perimeter set by the user or set by another approach (Figure 1).
- I suggest removing the titles in Figure 2 and give that information in the figure caption.
- In Table 1, the highest accuracy (94%) was estimated for Tubbs fire, but the AUC value (0.685 was one of the lowest), please check these values as they can be mistakenly written.
- There are some typing errors in the text that the authors should check (e.g., in the Discussion section the word "communities" was repeated; on page 15 "We used and modified the semi-physical model of Lautenberger (2017) and Purnomo et al. (2024)....").
- On page 14, it was stated that feature selection and feature engineering techniques were employed. If you applied feature selection, which method did you use? and which features or features did you eliminate after testing their relevancy?
- On page 16, F-beta was used to describe the popular accuracy metric. The most common term used for that metric is F-score or F-measure. I suggest you should check the current literature and use the most common term for the metric.
- On page 17, I suggest the revision of the text as "... and Gradient Boosting based XGBoost (Chen & Guestrin, 2016) since there is another method called Gradient Boosting Machine other than Extreme Gradient Boosting Machines (XGBoost).
- In the reference list, the scholar.google.com link is missing/empty page and inappropriate. Please revise the details of the reference.
- In the Supplementary file, please provide the accuracy results (similar to Table 1 in the manuscript) of logistic regression and random forest.

Version 1:

Reviewer comments:

Reviewer #1

(Remarks to the Author)

The manuscript titled " Fire Risk to Structures in California's Wildland-Urban Interface" intends to explore the critical factors influencing structure loss during wildland-urban interface (WUI) fires, particularly in California. The authors provide valuable insights into how home hardening, defensible space, exposure to flames and embers, and structure separation contribute to fire risk. By integrating machine learning models with exposure data, the study achieves a predictive accuracy of up to 82%, offering actionable data-driven strategies for fire mitigation.

The manuscript has been revised according to the review comments. The authors carefully studied the comments and revised the manuscript by considering all the comments.

In general, the manuscript is absolute complete compared to the previous one, since all the comments of the review have been revised. The Abstract, Methodology, Discussion and Conclusion sections are better, the number of references increased.

I believe the revised manuscript has been improved carefully and I hope the desired level of Journal of Nature Communications can be reached.

(Remarks on code availability)

Reviewer #3

(Remarks to the Author)

I am pleased to report that the authors have addressed all my revision requests in sufficient detail. The paper, with its new title and enhanced content, is now in excellent shape. I believe it will attract considerable attention and citations from the research community. The addition of new and revised tables, including the research results, has improved clarity and interpretability. In conclusion, I strongly recommend accepting the paper for publication.

(Remarks on code availability)

Reviewer comments:

Reviewer #1 (Remarks to the Author):

Thank you for the opportunity to review the manuscript titled "Isolating the Primary Drivers of Fire Risk to Structures in WUI Regions in California". This study explores the critical factors influencing structure loss during wildland-urban interface (WUI) fires, particularly in California. The authors provide valuable insights into how home hardening, defensible space, exposure to flames and embers, and structure separation contribute to fire risk. By integrating machine learning models with exposure data, the study achieves a predictive accuracy of up to 82%, offering actionable data-driven strategies for fire mitigation. While this research presents compelling findings, several areas require significant revision to align with the standards of Nature Communications and enhance the manuscript's clarity, rigor, and relevance.

Author comments - We thank the reviewer for their supportive comments and have extensively revised the manuscript as suggested.

The abstract is well-structured but should be refined to explicitly highlight the context, objectives, methods, results, and conclusions. A concise abstract is essential for effectively communicating the study's contributions to a broad readership. Additionally, the language should be made more engaging to encourage downloads and citations.

Author comments - We have updated the abstract to include more details while still keeping it concise, enhancing its clarity and readability as suggested.

The destructive impacts of wildfires on people, property and the environment have dramatically increased, especially in the Wildland-Urban Interface (WUI) in California. In these areas structures are threatened by both approaching flames and lofted embers which spread fire into and within communities. While independent factors influencing structure fire protection are well known, their combined effects remain largely unquantified, limiting the accuracy of risk assessments and mitigation strategies. Here, we examine five major historical WUI fires—2017 Tubbs, 2017 Thomas, 2018 Camp, 2019 Kincade, and 2020 Glass Fires—utilizing machine learning (ML) analysis of on-the-ground post-fire data collection, remotely sensed data, and fire reconstruction modeling to assess patterns of structure loss and mitigation effectiveness. We show that the spacing between structures is a critical factor influencing fire risk, highlighting the importance of structure density, while fire exposure, the ignition resistance (hardening) of structures, and clearing around structures (defensible space) work in combination to mediate fire risk. Utilizing an XGBoost ML model, structure survivability can be predicted to 82% accuracy. Results highlight the effectiveness of hardening and defensible space, with a hypothetical 52% reduction in losses. Our findings emphasize the need for community-level mitigation to reduce structure loss in future WUI fires.

A well-defined research problem is crucial for a strong manuscript, and while this study addresses an important issue, the problem statement could be more clearly articulated. The introduction should explicitly state the gap in existing research and how this study fills that gap. Clearly outlining the research questions and hypotheses will provide readers with a compelling rationale for the study and its broader significance.

Author comments - We have restructured our introduction to more clearly include the gaps in current research as a defined paragraph, our hypotheses and the problem statement in the last two paragraphs of the introduction. We copied these below and highlighted these sentences in **Bold**.

Despite these advances, the majority of past studies focus on single events, and lack a comprehensive quantitative analysis of how mitigation measures, such as home hardening and defensible space, interact and influence fire risk. In order to safeguard communities and stem the current trend of destruction, we must quantitatively understand how features influence fire risk to structures, particularly in relation to fire exposure, surrounding vegetation, the proximity of neighboring structures, and properties of the structures themselves. We hypothesize that the combined effects of structure hardening, defensible space, and structural separation can significantly reduce the risk of structure loss, with the most substantial benefits occurring when changes are made to both the structure itself and the surrounding vegetation. Furthermore, changes to individual structures may not be sufficient to reduce risk when structures are arranged at high density, requiring community-wide mitigation.

Here, we combine the largest existing structure loss database from California with simulated fire and ember exposure conditions across multiple large-loss events, providing a methodology to quantify and compare the combined influence of exposure and mitigations such as defensible space and home hardening on fire risk. Unlike past studies, fire reconstruction modeling that includes urban fire spread is used to quantitatively estimate the effect of flame and ember exposure on structures. Geospatial assessments of vegetation surrounding structures are added using both LiDAR and visual imagery to assess the level of defensible space (vegetation) surrounding structures. The database is then fit using a multivariate analysis similar to Mockrin et al. (2023) and Metz et al. (2024) that distinguishes between the interrelated effects of exposure, structure hardening, and defensible space. A parameter importance analysis reveals the strong role both structure separation and exposure play, distinguishing wildfire from other natural hazards that are not affected by neighboring conditions, highlighting the importance of a community approach to mitigation. The model developed is strongly predictive when incorporating all the above features and is also used to assess the impact of recommended mitigation measures on homes. It is found that it is necessary to make changes both to the structure itself and surrounding vegetation, especially that closest vegetation within 1.5 m (5 ft) of the structure (zone 0) to achieve the maximum benefit.

The discussion section should include a more thorough examination of the study's limitations. For example, the reliance on post-fire damage assessments, potential biases in the dataset, and limitations of the machine learning approach should be acknowledged. Furthermore, the manuscript would benefit from a stronger discussion of policy implications. Given the increasing severity of wildfires in WUI regions, how can these findings inform fire safety regulations, urban planning, or emergency response strategies? Adding specific recommendations will significantly enhance the real-world applicability of the research.

Author comments - An extensive description of the limitations of input data, numerical modeling, and machine learning models have been added to the end of the discussion section as follows:

It is important to highlight potential limitations of the input data and methods used in this study. While reliance on post-fire damage inspection (DINS) assessments provides clear distinctions on the damage state of structures, there is inherent uncertainty on collected features that are difficult to determine in a post-damage state. Additionally, the DINS data may not capture the full range of potential factors that contribute to fire damage, such as local preparedness, fire suppression efforts, construction practices, etc., which could affect the generalizability of the findings to other regions or future fires. Remotely-sensed data, particularly for vegetation separation distance (VSD) is also subject to potential errors as small fuels that can contribute to local fire spread may not be visible at the resolution captured. The moisture content, species, and arrangement of vegetation and other flammable materials may also influence the effectiveness of defensible space but can't be captured through remote sensing modalities. Estimates of fire and ember exposure from fire reconstruction modeling are also subject to potential errors from deviations in fuel or weather input data and the empirical nature of the model itself. Machine Learning models are also limited in their reliance only on provided data. Efforts have been taken to minimize these potential sources of error, including conducting a comprehensive sensitivity analysis with perturbations, an assessment of the trained ML model to systematically assess how variations in key parameters influence our predictions, and use of a validation dataset to assess the performance of the model.

The manuscript presents noteworthy results. The finding that structure separation and exposure to embers and flames are the strongest predictors of structure loss is particularly significant. The results highlight the crucial role of urban conflagrations in fire spread, distinguishing them from smaller wildland fires. The study also provides compelling evidence that home hardening and defensible space—particularly within five feet of a structure—can reduce hypothetical structure losses by 52%. This reinforces the importance of mitigation strategies tailored to WUI communities.

Author comments - We appreciate the reviewer's comment.

The study is highly relevant to wildfire risk management and has potential applications in related fields such as urban planning, forestry, and disaster resilience. When compared to existing

literature, the research aligns with previous studies emphasizing the role of home hardening and defensible space. However, some findings, particularly regarding the dominance of structure separation over other factors, challenge traditional assumptions. Expanding the literature review to provide more context for this finding would strengthen the discussion.

Author comments - We have included several recent studies to expand the literature and provide a more comprehensive understanding of wildfire resilience and mitigation strategies. We copied these below and highlighted these studies in **Bold**.

Central to preventing future destruction has been the development of mitigation measures aimed at reducing the likelihood of ignition and spread in the WUI (Calkin et al., 2014; Calkin et al., 2023; Mahmoud, 2024; Naser & Kodur, 2025; Pandey et al., 2023). Improvements in building features and materials (hardening and clearing surrounding vegetation and other flammable materials (defensible space) play important roles mitigating fire spread into the WUI (Cary et al., 2009; Cohen, Jack D, 2000; Maranghides & Mell, 2013; Quarles et al., 2010) but differ in their characteristics because structures and vegetation have different heat release rates, durations of burning, and responses to external exposure including direct flame contact, radiation, and firebrands (Caton et al., 2017). For instance, Ondeï et al. (2024) synthesize a zonation strategy for defensible space, focusing on removing dead vegetation within 1.5 meters of a house and managing fuel connectivity up to 30 meters. Similarly, studies like Carton et al. (2024) stress the importance of fire-resistant construction, vegetation management, and the need for specific wildfire codes, particularly addressing the unique needs of Indigenous communities and heritage properties in Canada. While effective mitigation strategies have been developed based on past testing and investigations (Maranghides et al., 2022), their combined effectiveness under different exposure conditions is not yet known (Schoennagel et al., 2017).

*Previous geospatial studies have demonstrated the critical influence of spatial arrangement and biophysical factors (Syphard et al., 2012, 2017; Alexandre et al., 2016), with defensible space around structures playing a significant, albeit secondary, role (Syphard 2014; Mockrin et al., 2023). The role of building materials has also been examined, revealing mixed findings (Mockrin et al., 2023; A. Syphard & Keeley, 2019; Troy et al., 2022). For instance, Syphard & Keeley (2019) found structural features like enclosed eaves and vent screens were crucial, while others (Price et al., 2021, Metz et al., 2024, Knapp et al., 2021) identified factors such as spacing and arrangement as more significant, suggesting determinants of loss are often beyond homeowners' control. **A later study of the Woolsey Fire suggests that proximity to destroyed structures and building materials, such as multi-pane windows and enclosed eaves, are key factors in determining survival (Mockrin et al., 2023). Large structure loss datasets, such as those from the Camp fire, show that homes built before 1997 had significantly lower survival rates compared to those built after, underscoring the importance of construction standards (Knapp et al., 2021). The 2021 Marshall***

Fire also highlights the significance of neighborhood and parcel characteristics in housing survival, revealing the impact of jurisdictional differences in building codes and planning (Metz et al., 2024). Collectively, these studies underscore that while defensible space is important, building features and surrounding vegetation, as well as proactive mitigation strategies, are critical to improving wildfire resilience.

While the data analysis is robust, the study could benefit from additional validation. The machine learning models used are effective, but the conclusions regarding mitigation strategies should be carefully framed to acknowledge that real-world application may vary. Additional case studies or sensitivity analyses would help confirm the generalizability of the results.

Author comments - Validation of the machine learning models was conducted by splitting the dataset into training and testing subsets. The training data was used to fit and optimize the models, while the testing data served to evaluate their generalization performance. This approach assesses how well each model can predict unseen data and ensure that it is not overfitting to the training set. Furthermore, we employed cross-validation techniques to further validate the models' performance, providing a more robust evaluation by testing them on different subsets of the data. This comprehensive validation process helped ensure the reliability and accuracy of the models before deployment.

We also developed a comprehensive sensitivity analysis framework to assess how variability in key input features from the exposure model (ember load and flame length) affects our model predictions. The plots and metrics added to the method section. Regarding the interpretation of machine learning models, we used the SHAP method. SHAP can be considered a form of sensitivity analysis because it assesses how changing a feature or a subset of features affects the model's output. It evaluates the impact of including or excluding a feature and identifies which features contribute most to the predictions. However, SHAP differs from traditional sensitivity analysis in several ways. Traditional methods typically involve perturbing feature values continuously, and observing the resulting changes in output. In contrast, SHAP is based on cooperative game theory and evaluates all possible combinations of features to determine their individual contributions. Additionally, SHAP provides both local explanations (for individual predictions) and global insights (by aggregating feature importance across all instances), offering a more comprehensive understanding than standard sensitivity techniques. In summary, we covered both traditional sensitivity analysis as well as the SHAP method to assess and interpret the impact of input features on model predictions. The traditional sensitivity analysis helped quantify the uncertainty and variability in the model's responses to changes in ember load and flame length, while SHAP values provided deeper insights into the contributions of individual features. By combining these approaches, we gained a comprehensive understanding of the model's behavior, identifying key drivers of model output and revealing potential areas of improvement. This dual approach not only enhanced the interpretability of the model but also allowed for more informed decision-making in mitigating wildfire-related risks.

For the mitigation strategies, we employed hypothetical scenarios derived from existing literature and engineering intuitions to estimate the effectiveness of various measures and their interactions. We have provided further clarification on this approach and highlighted the potential differences when applied to real-world situations, including in the discussion and throughout the methods section, including a new section on “Sensitivity Analysis for the Machine Learning Model.”

The methodology is sound and meets expected standards in the field, particularly in the use of geospatial assessments, machine learning techniques, and predictive modeling. However, more detail is needed to ensure full reproducibility. The manuscript should provide clearer descriptions of data preprocessing steps, variable selection criteria, and model training procedures. Transparency in these areas will allow other researchers to replicate and build upon the findings.

Author comments - The Methods section has been thoroughly revised to include additional details on pre-processing, machine learning techniques and classifiers, as well as the SHAP method. For the pre-processing steps, we specify the criteria and methods applied to each feature in the data preprocessing pipeline, ensuring that each variable is appropriately prepared for model training. This includes handling missing values, encoding categorical variables, scaling numerical features, and addressing any class imbalance. The choice of machine learning techniques and classifiers is detailed, with justifications for selecting each method based on the nature of the data and the modeling objectives. Furthermore, the SHAP method is outlined to explain the model’s decision-making process, offering a transparent view of feature contribution effects.

In addition to detailed workflow and data processing described in this work, the paper includes links to a repository and data that should make it easier to reproduce the results of this study.

In addition to these points, a few formatting and structural improvements are needed. The journal recommends keeping the main text (Introduction, Results, and Discussion) under 5,000 words and limiting the number of figures and tables to 10. Ensuring adherence to these guidelines will improve the manuscript’s chances of acceptance. The reference list should also be expanded to include 60–70 citations, prioritizing recent and globally relevant studies to strengthen the paper’s scholarly foundation.

This manuscript offers valuable insights into wildfire risk and mitigation, making it a strong candidate for publication. However, significant revisions are required to refine the abstract, enhance the discussion of policy implications, address limitations, and ensure methodological clarity.

Author comments - We have updated and revised the manuscript to comply with all noted journal guidelines, including expansion of the text, inclusion of additional recent and globally-relevant studies, and keeping the text and figure count below the limits. We have also included an extensive and explicit discussion of policy implications in the discussion section.

Reviewer #2 (Remarks to the Author):

Hello! It was a pleasure to read your paper. It's good to see new research and methods reinforce previous results. The writing is clear, concise, and effective.

Noteworthy results are that:

- Several factors are at play during a fire event that affect the outcome of structure loss (previous research had similar conclusions, but this is a new approach and reinforces previous results).
- there is still opportunity to mitigate fire risk.
- Community-wide actions are important.
- Structure Separation Distance is a key metric when characterizing likelihood of loss.

The methods are sound as far as my understanding and there is enough detail in the methods section for the work to be reproduced.

Minor correction(s): I noticed that one of your references is missing from the reference's list (Mockrin et al., 2023); This is biased on my end because I was the author, but on the first paragraph of the Discussion (lines 2/3)), my research from 2016 also said that the spatial arrangement of building plays an important role. That paper is already in the reference list, but I confess I feel proud to see other researchers, with different methods, came to similar conclusions. If you agree with my take, would you be willing to add my paper "The relative impacts of vegetation, topography and spatial arrangement on building loss to wildfires in case studies of California and Colorado"?

Author comments - We appreciate the reviewer's comment. We revised the reference list and we also added the following paper to the Discussion section:
Alexandre, P. M., Stewart, S. I., Mockrin, M. H., Keuler, N. S., Syphard, A. D., Bar-Massada, A., Clayton, M. K., & Radeloff, V. C. (2016). The relative impacts of vegetation, topography and spatial arrangement on building loss to wildfires in case studies of California and Colorado. *Landscape Ecology*, 31(2), 415–430.

I recommend publishing this paper.

Thank you,
Patricia Alexandre.

Reviewer #3 (Remarks to the Author):

The effect of the Wildland-Urban Interface (WUI) was investigated in this study using extensive datasets. The authors combined California's largest existing structure loss database with simulated fire and ember exposure conditions to structures across multiple large-loss events,

comparing the effects of structure hardening and nearby defensible space. Fire reconstruction modeling including urban fire spread was implemented using Eulerian Level Set Model of FIRE (ELMFIRE). Three machine learning algorithms (CatBoost algorithm was mentioned on page 10, but nothing was mentioned in the Methods and Results section. Therefore, it was ignored in my report), including logistic regression, random forest, and XGBoost algorithms. Logistic regression method is one of the simplest methods applied in the literature as a benchmark method, but its performance is usually poorer than advanced machine learning and deep learning algorithms. Three scenarios were conducted to measure the effect of mitigation measures and it reported that a significant reduction in the loss can be achieved with these scenarios. I think the paper has some novelty and may deserve to be published. However, there are some issues that the authors must consider to revise in terms of technical soundness and compliance with journal style.

Author comments - We thank the reviewer for their helpful comments. We first note that CatBoost was not originally included in the main text (only the methods section), but we have now updated the manuscript to explicitly list the CatBoost model in the main text as well.

In the CatBoost model, the SHAP values shown in the figure below (for the WUI data and Individual Fires) are derived from a binary classification setup (Class 0 = Survived, Class 1 = Destroyed). The training and testing for this algorithm typically require minimal preprocessing because CatBoost is designed to natively handle categorical features without extensive one-hot encoding, which simplifies the pipeline and preserves the original feature relationships.

In contrast, when using Logistic Regression, Random Forest, or XGBoost, additional preprocessing steps are often necessary. For example, logistic regression usually requires scaling or normalization of numerical features and converting categorical variables into dummy variables. Random forest and XGBoost, while also capable of handling a mix of feature types, generally benefit from explicit encoding of categorical features (such as one-hot encoding) to ensure that the model can properly interpret the feature values. This additional step increases the dimensionality of the dataset, potentially leading to a more complex model tuning process and longer training times.

In summary, while CatBoost can seamlessly integrate categorical data with minimal preprocessing and achieve competitive performance on binary classification tasks like ours, logistic regression, random forest, and XGBoost typically require more elaborate feature engineering and preprocessing, which in turn can influence both model performance and the interpretability of sensitivity analyses such as those based on SHAP values. Therefore, CatBoost is now added to the results section but a full description of the methodology used is given in the methods section and supplementary materials, rather than in the main body of this paper to conform with space limitations.

-Random forest has been a popular ML algorithm, and it is used in almost all fields, with high level of performance reported. One of the most superior characteristics of the Random Forest algorithm is that it can be directly used for feature importance estimation using its Mean Decrease Impurity (MDI) or Gini importance. In this study, the performance of the random forest method was not provided. In the first paragraph of the Damage Prediction Results section, prediction performances of logistic regression and XGBoost were given as estimated accuracies

of 78% and 82%. In the next paragraph of that section, although the statement "A comparative analysis between the Logistic Regression, Random Forest, XGBoost and CatBoost models is included in the methods section, but importantly underscores the need for selecting an appropriate algorithm based on the specific characteristics of the dataset and outcome to be achieved" was given, no result was provided for random forest (also CatBoost as mentioned above) in the main document and the supplementary document. This is one of the main results that must have been given, and a comparative discussion should be provided. The authors should discuss the similarities and differences between the SHAP graph of Random Forest and its Gini index results (for the entire datasets as given in Figure 4). SHAP is a model-agnostic approach and can be applied to any method. However, feature importances of random forest are automatically produced by the model. SHAP findings for feature importances may be different, which requires a short discussion (this is important to prove the trustworthiness of the SHAP method as an explainability model).

Author comments - We appreciate the reviewer's insights about the Random Forest method. We acknowledge that while the Gini index (or Mean Decrease Impurity) is a widely used measure in Random Forests for feature importance estimation, it comes with known limitations such as a bias toward features with many levels or high cardinality. In our study, we opted for SHAP (SHapley Additive exPlanations) as a model-agnostic tool to interpret our models because SHAP values not only quantify the magnitude and direction of each feature's contribution but also provide a more nuanced view of feature interactions and non-linear effects. Although each model—Logistic Regression, Random Forest, XGBoost, and CatBoost—offers unique advantages, our final model selection was guided by overall prediction accuracy, model complexity, and the efficiency of hyperparameter tuning on our dataset.

For the purpose of comparison, we utilized the Random Forest model with the Gini index to assess and display the feature importance across the entire dataset. The results of this analysis are presented as follows:

To enhance the ranking and evaluation of the models and their prediction metrics, we have included a detailed table (Table 1) in the main text that provides a clear overview of each model utilized in our study. This table presents key metrics, such as Accuracy, AUC, and the most

significant features for each model, specifically tailored to the DINS data from our study period (2017-2022). Additionally, the table includes the results derived from combining data from all five fires analyzed, offering a comprehensive comparison of the models' performance across different scenarios. This approach allows for a more transparent and comparative assessment of how each model contributes to the overall prediction accuracy and feature relevance.

In the supplementary materials, we have provided a comparative analysis that includes a detailed table showcasing the prediction metrics for Logistic Regression, Random Forest, and CatBoost models. This table offers a comparison of key performance indicators, such as accuracy, AUC, and other relevant metrics. Additionally, we have expanded on our rationale for selecting XGBoost as part of our modeling approach, highlighting the model's overall performance across various evaluation criteria. This explanation emphasizes XGBoost's ability to consistently deliver strong results, particularly in handling complex data relationships and providing robust predictions. The supplementary materials aim to offer a clearer understanding of the models' relative strengths and the factors influencing our model selection process.

A similar problem exists in the interpretation of the logistic regression results in that SSD, which is the most important feature in the XGBoost model, was the least effective feature. The accuracy difference was about 4%, and such a difference was reported in feature importances. Please make some comments on the results, by giving possible reasons.

Author comments - One possible explanation is that the XGBoost model, as a tree-based ensemble, can capture non-linear interactions and threshold effects between SSD and other variables. In our dataset, Structure Separation Distance (SSD) may interact with other features in a complex, non-linear manner that significantly improves predictions when modeled by XGBoost. On the other hand, logistic regression assumes a linear relationship between the features and the log-odds of the outcome. As a result, it may fail to capture these non-linearities and interactions, leading to a lower apparent importance for SSD and a roughly 4% accuracy difference. This discrepancy underscores how the complexity and non-linearity in the data can favor more flexible models like XGBoost over simpler linear models when it comes to feature importance estimation and overall predictive performance.

- In the title instead of using the term "Isolating", I suggest the authors should prefer another term (e.g., Determining, Uncovering, Identifying).

Author comments - We have updated the title to avoid the term "isolating" and also make it more concise to conform with the broader style of Nature Communications. It is now "Fire Risk to Structures in California's Wildland-Urban Interface".

- Authors should mention the name of the methods and the scenarios in the abstract section. In its current form, it is too generic, not including the details of the study and important results achieved in this study.

Author comments - The abstract has been updated to provide more details.

-Please follow the instructions for authors on how to cite articles in the main text (numbering) and section heading. I think the section called Main should be revised as Introduction.

Author comments - The reference section and manuscript format have been adjusted according to journal formatting guidelines.

- I am confused about the datasets, more specifically the features used in the analysis. On page 1 (where the authors start to explain the data in their study)", it was stated that " Geospatial assessments of vegetation surrounding structures are added using both LiDAR and visual imagery to assess the level of defensible space (vegetation) surrounding structures." Did you use LiDAR and visual imagery data in this study to prepare your dataset (in SHAP graphs there are totally 10 features or factors considered)? Please clarify this issue.

Author comments - LiDAR was used where available in order to calculate vegetation separation distance (VSD). In counties where recent LiDAR data was not available, we used visual imagery to create buffers for VSD. Because we needed data before the fire occurred, we had to use different datasets depending on timing and availability. Generally, these data are aligned when compared to visual measurements. We established three zones based on CAL FIRE recommendations (Zone 0 = the first 5 feet, Zone 1 = within 30 feet, Zone 2 = within 100 feet) between structures and surrounding vegetation, using vegetation density in these zones to represent the influence of defensible space (which is measured as VSD in this study).

- Please give some information on the fire perimeter set by the user or set by another approach (Figure 1).

Author comments - The final fire perimeter in this study used from CAL FIRE database as well as observed fire perimeters (from NIFC) considering a 300 ft (91m) buffer to include affected structures as well as undamaged structures.

- I suggest removing the titles in Figure 2 and give that information in the figure caption.

Author comments - We added a caption to the figure and removed the titles.

- In Table 1, the highest accuracy (94%) was estimated for Tubbs fire, but the AUC value (0.685 was one of the lowest), please check these values as they can be mistakenly written.

Author comments -The low number of the AUC value in comparison to Accuracy is because the accuracy metric is based on a threshold, typically 0.5, but AUC considers all possible thresholds (from 0 to 1) for classification. Since the accuracy is high, it implies that the model is predicting most of the positives correctly, but it could be biased towards the positive class. We added Precision, Recall, and F1-score metrics that represent better evaluation of the model.

- There are some typing errors in the text that the authors should check (e.g., in the Discussion section the word "communities" was repeated; on page 15 "We used and modified the semi-physical model of Lautenberger (2017) and Purnomo et al. (2024)...").

Author comments - We revised these errors, thank you.

-On page 14, it was stated that feature selection and feature engineering techniques were employed. If you applied feature selection, which method did you use? and which features or features did you eliminate after testing their relevancy?

Author comments - In our preprocessing pipeline, feature selection was integrated within the feature engineering and data preparation steps. While no explicit standalone feature selection method was applied, we performed several operations that effectively reduced and transformed the feature set. This included dropping features with low relevance or high redundancy, such as "Zip Code," and "Latitude" and "Longitude" after converting them to UTM coordinates. Additionally, the pairwise distance imputation and Haversine imputation methods helped deal with missing data by considering geographic location, further refining the data.

Feature engineering steps such as creating new features (e.g., SSD from "Distance" utm_easting/utm_northing from latitude and longitude) ensured that only the most relevant information was retained for the models. Additionally, categorical features were encoded with OneHotEncoder, and numerical features were standardized with StandardScaler. After these steps, a set of numerical and categorical features were retained for model training, and some columns like "Latitude," "Longitude," "utm_easting," "utm_northing," and others were dropped.

Thus, feature selection was indirectly applied by removing irrelevant or less useful features during preprocessing, ensuring that the model focused on the most meaningful variables.

-On page 16, F-beta was used to describe the popular accuracy metric. The most common term used for that metric is F-score or F-measure. I suggest you should check the current literature and use the most common term for the metric.

Author comments - Thank you for the reviewer's comment. It is correct that the term "F-score" or "F-measure" is widely used in the literature. However, in our study, we specifically used the F-beta score for hyperparameter selection, which is a variant of the F-score that allows for adjusting the balance between precision and recall through the beta parameter. This makes F-beta the appropriate term in this context, as it gives us the flexibility to tune the model's sensitivity to false positives versus false negatives.

We have also added the F1-Score to Table 2, so we used both F-beta and F1-score for hyperparameter tuning and model performance evaluation. We will ensure that the explanation in the manuscript makes it clear that we are using F-beta score for hyperparameter selection.

Hyper parameter selection is performed using the best result in terms of the following classification metrics: **F-beta, F1-Score, accuracy, balanced accuracy and precision-recall scores. The F-beta score is used to balance precision and recall, with the beta parameter allowing for tuning the model's sensitivity to false positives and false negatives.**

- On page 17, I suggest the revision of the text as "... and Gradient Boosting based XGBoost (Chen & Guestrin, 2016) since there is another method called Gradient Boosting Machine other than Extreme Gradient Boosting Machines (XGBoost).

Author comments - We have revised this statement for better clarification on the XGBoost method.

- In the reference list, the scholar.google.com link is missing/empty page and inappropriate. Please revise the details of the reference.

Author comments - We modified the reference list to make sure all links are included.

- In the Supplementary file, please provide the accuracy results (similar to Table 1 in the manuscript) of logistic regression and random forest.

Author comments - We have added the tables of metrics for Logistic Regression, Random Forest, and CatBoost (Tables 1-3) to the supplementary materials.